# Forensic Analysis Laboratory for Sport Devices: A Practical Use Case

Pablo Donaire-Calleja, Antonio Robles-Gómez *, Llanos Tobarra and Rafael Pastor-Vargas

Departamento de Sistemas de Comunicación y Control, Escuela Técnica Superior de Ingeniería Informática, Universidad Nacional de Educación a Distancia (UNED), 28040 Madrid, Spain; pdonaire7@alumno.uned.es (P.D.-C.); llanos@scc.uned.es (L.T.); rpastor@scc.uned.es (R.P.-V.)
* Correspondence: arobles@scc.uned.es

**Abstract:** At present, the mobile device sector is experiencing significant growth. In particular, wearable devices have become a common element in society. This fact implies that users unconsciously accept the constant dynamic collection of private data about their habits and behaviours. Therefore, this work focuses on highlighting and analysing some of the main issues that forensic analysts face in this sector, such as the lack of standard procedures for analysis and the common use of private protocols for data communication. Thus, it is almost impossible for a digital forensic specialist to fully specialize in the context of wearables, such as smartwatches for sports activities. With the aim of highlighting these problems, a complete forensic analysis laboratory for such sports devices is described in this paper. We selected a smartwatch belonging to the Garmin Forerunner Series, due to its great popularity. Through an analysis, its strengths and weaknesses in terms of data protection are described. We also analyse how companies are increasingly taking personal data privacy into consideration, in order to minimize unwanted information leaks. Finally, a set of initial security recommendations for the use of these kinds of devices are provided to the reader.

**Keywords:** forensic analysis; sport devices; data privacy; Internet of Things (IoT); virtual laboratories; security recommendations

## 1. Introduction

The revolution of the Internet in combination with emerging technologies is transforming our daily lives. The Internet is a global network including various nodes or devices which are capable of communication. These nodes interact across heterogeneous hardware and software platforms. In this line, the Internet of Things (IoT) [1] paradigm includes a great variety of specific technologies, communications protocols, smart devices, and so on. As such, it has become a significant technology posing great challenges in the digital and industrial fields, such as e-health [1], agriculture [2], and smart cities [3], among others. The cybersecurity issue is very present in these kinds of IoT devices [4,5].

The forensic analysis discipline regarding wearable devices employed for sports is a particular field of interest in the context of IoT. In particular, smartwatches and fitness devices are the most popular among the different wearable devices [6], due to their great mobility and connectivity capabilities [7]. In this way, they are continuously connected to users' mobile devices with a large variety of sensors and specific components, including microphones, GPS, accelerometers, cameras, etc. They also offer notifications, alerts, recommendations, among other utilities [8,9]. According to multiple studies carried out after global lockdown [10], there has been a big increase in the number of physical activities carried out by people. The principal motivations for carrying out these habits are becoming in good physical shape, and medical reasons, among other reasons. Specifically, a recent study conducted by the Spanish Ministry of Culture and Sports [11], 6 out of 10 (i.e., 57.3%) Spanish people older than 15 years participate in sports, either periodically or occasionally.

This value represents an increase of 3.8% compared to 2015 when the percentage stood at 53.5%.

These reasons and the social motivation caused by different mobile applications, such as Strava, Runtastic, or StepBet, among others, have increased the sales of sport accessories in recent years [12]. However, although Internet users are concerned about their data entering the network—which has a great economic value and is something to be concerned about—they usually do not know which data are being captured by these applications or the associated electronic devices. More than 66% of runners use wearable devices to quantify their sport performance [13]. Specifically, Garmin has been found to be the most popular brand among users of sports devices with almost 44% of runners, whereas Polar, TomTom, and Nike are used less [14]. Garmin is one of the main companies in the sector. In addition, the roles that sports trackers, including several kinds of Garmin devices, and running-related data play in runners' personal goal achievement are explored in [15]. Runners are very motivated by documenting and tracking their activities, as well as supporting their goal-oriented reflections and actions.

Therefore, in this work, we first detail how Garmin guarantees the data security of their devices, showing how sensitive data such as WiFi connections or Bluetooth pairings are kept encrypted. It is also left in the hands of the owners to keep information concerning their training locations. This information, as will be shown in this work, is fully exposed and decrypted, and can be acquired to generate an activity map or even trace user locations. Although users understand that this data may be available, they should be more aware that it can be recovered using forensic techniques, even if it has been manually removed from the device.

In addition to this, in-depth research of the data collected by a common sport smartwatch belonging to the Garmin Forerunner Series, with different planned workout activities, is performed. On one hand, we determine which information is kept in the device. On the other hand, sensitive data could be exposed during a digital forensic analysis or captured by malicious people who could gain access to it. The actions necessary to carry out these phases require modifying the current paradigm, due to the lack of software tools prepared for these devices or the lack of standards at both the hardware and architecture levels. The guidelines of the UNE 71506:2013 [16] and UNE-EN ISO/IEC 27037:2016 [17] standards were followed as a starting point in this work.

This work makes the following contributions:

- Proposing a specific ecosystem (a virtual laboratory and associated tools) for the digital forensic analysis of sport devices.
- Analysing a practical case study with a real sport device by detailing the different phases of a classic methodology of digital forensic analysis, as well as considering several standards.
- Giving some security guidelines to preserve the information that users share with sport devices, consequently guaranteeing the data privacy of users.

The remainder of this paper is organized as follows. Section 2 reviews the state-of-the-art techniques in the context of forensic analysis for sport devices. The methodology of our proposal and the creation and configurations of the designed work environment are detailed in Section 3. Our proposed solution for the forensic analysis of sport devices is given in Section 4. Section 5 describes the obtained results and provides a set of recommendations for these kinds of sport devices. Finally, some conclusions and directions for further work are outlined in Section 6.

## 2. Related Work

According to the existing literature, some earlier works have carried out digital forensic investigations of various sport devices. Due to the great variety of these devices, each requires specific procedures and many of them are composed of proprietary binary files that make it hard to extract any useful information. Dawson et al. [18] performed an in-depth analysis of the Tom Tom Spark 3 smartwatch. The authors emphasized that it is mandatory

to use tools (mostly open source) that fall outside the classic standards of forensic analysis, or even which are not widely approved due to the lack of a clear consensus among the scientific community. Despite the fact that the user can remove traces, the device still contains location data including time and place indicators.

MacDermott et al. [19] studied three smartwatches: A Garmin, a Fitbit, and a generic fitness tracker. In their analysis, they emphasized how current tools for classic forensic analysis are inadequate for a complete study and the need for open-source tools to visualize file information. In addition, they stated that the big problem is the disparity of brands and models, as well as the lack of standard procedures, as a different method was needed for each analysed device. Yoon and Karabiyik [20] carried out a forensic analysis on a Fitbit Versa 2 sport watch, a widely accepted model. Their work revealed that sensitive documents, such as certain bank information, can be obtained for malicious purposes. At Johns Hopkins University [21], research was also performed on a Fitbit sport smartwatch; however, they demonstrated the viability of performing a complete forensic analysis without the use of commercial tools.

A research team from Adelaide University [22] carried out an investigation in 2016 that clearly exposed privacy risks for the user of sport devices, discovering how much the companies (or a person with malicious purpose) can learn about a particular user from this information. Their study focused on a Samsung Gear Live device with Android OS, revealing deficiencies in terms of data protection security and showing possible attack vectors for information leaks. A man-in-the-middle scenario was also emulated for several commercial fitness devices to examine data privacy transmitted by each application tested in [9]. To date, there have not been many works regarding the methodologies used for the appraisal of these devices, due to the great difficulty of proposing a standard for the variety of architectures. Jawad Abbas [23] performed a comparative review of the different frameworks of forensic assessment over the years and their evolution.

On the other hand, a promising digital forensic analysis of several wearable devices was comparatively performed in [24]. In particular, a set of Samsung Galaxy and Apple smartwatches, as well as the Garmin Vivosport smartband, from the logical and physical points of view. In our case, our work looks to build on previous works to demonstrate the forensic analysis of one of the most common devices in the sector, the Garmin smartwatch. Its strengths and weaknesses, in terms of user data protection—both on the device and during the exchange of information with the Garmin cloud service—are analysed. For this purpose, a classic procedure is followed in this study, relying on open-source tools that are widely used by the community whenever possible. The proposal also has a formative purpose: the creation of a virtual environment, which can be reproduced in the context of forensic analysis, as an educational laboratory.

Specifically, the experiments of our research study are based on the specific Garmin Forerunner 920XT smartwatch in the context of forensic analysis without any physical manipulation during the whole forensic analysis process. Physical forensics (chip-off approaches) could damage the device, so it is not a repeatable procedure by a third party. In addition, from a judicial point of view, the authorization of physical forensics is very rare in our context. Thus, since this smartwatch is widely employed by many athletes, it is necessary to carry out a study by emulating a digital forensic analysis process taking into account these circumstances. As our approach is under a formative perspective, the most common approach is described. Additionally, our study goes deeper by examining and analysing possible vulnerabilities within the interactions among this device and the associated cloud services of Garmin, such as user's activities, data storage, etc. We consider the protection of user data obtained from such devices as a very relevant feature in this work.

## 3. Material and Methods

### 3.1. Methodology

A deep study was performed of a smartwatch belonging to the Garmin Forerunner Series family, prior to its synchronization with the associated Garmin cloud service. In this

case, the IoT device will communicate with the cloud through a WiFi connection. The IoT and cloud paradigms are highly related. The monitoring of this synchronization process is also analysed, considering data possibly vulnerable to attack vectors such as man-in-the-middle attacks [9]. Finally, additional forensic analysis was carried out to determine the data remaining after the uploading process.

In accordance, a methodology validated and accepted by the community should be followed for a suitable digital forensic analysis. However, as there is currently no standardized procedure for these types of devices, a classic methodology was followed, as well as specific tools to undertake concrete actions in which elements of these devices are involved. In Europe, two standards can be found: (1) UNE 71506:2013 [16], which establishes a methodology for preservation, acquisition, analysis, documentation, and presentation; and (2) UNE-EN ISO/IEC 27037:2016 [17], which contains guidelines for the acquisition phase.

These standards have been adapted for our purposes, following the phases detailed below:

- Preparation. The creation and configuration of different environments (or virtual laboratories) to conduct the forensic analysis are carried out. The designed practical case study is suitable for both a forensic analyst and formative training. The virtual environment specifically designed for this work can be replicated easily.
- Acquisition. The user's personal data is collected, in order to be analysed and documented in the next phases. Specific tools used for the data collection process are detailed.
- Documentation. The documentation of each digital evidence obtained from personal data is stored in the Autopsy software.
- Analysis. This task of data analysis is merged with the documentation phase for formative purposes. The specific tools used for all the analyses performed in this work are also detailed.
- Presentation. The conclusions obtained after performing the forensic analysis are presented and reported. Furthermore, some security guidelines for these specific sport devices in the context of IoT are exposed as recommendations.

### 3.2. The Virtual Environment and Tools

The designed environment for this work can be seen as a virtual laboratory, which is composed of several elements. Oracle Virtual Machine (VM) VirtualBox virtualization software is installed in our host machine as a hypervisor; however, a different OS and virtualization software could be employed. Figure 1 shows the architecture of this virtual environment, as follows:

- A forensic setup, which is made up of two VMs is incorporated into the virtual environment. One of them includes specific Linux-related tools, and the another one specific Windows tools. The main reason for using two forensic VMs is to facilitate the use of graphical tools, such as FTK Imager. These are:
  1. The Linux VM is pre-loaded with the Parrot OS [25], with forensic applications dedicated to network-monitoring analysis. This distribution is focused on cybersecurity purposes, including many tools for offensive teams (Red Team) and defensive teams (Blue Team), which are already installed and configured. This VM is isolated from the Internet during the analysis phase.
  2. The Windows VM is equipped with the rest of the forensic applications, which is used during the different phases of the analysis.
- A user setup is included in the virtual environment by means of a user VM. It only has the Garmin Express client application installed, in order to connect with the Garmin cloud service. Windows is installed in this VM. It plays the role of the user and all the user and communication actions are monitored with the Parrot VM. In this way, it can be guaranteed that there are no elements that could influence the analysis results. The

OS type is provided in accordance with the reviewed forensic best practices to enable the replication of each analysis phase. Other alternatives could have been employed for both memory and network analyses.

The virtual laboratory also contains VMs created from scratch, specifically for the forensic analysis process. In our case, this is the user VM. Therefore, there is a VM to simulate a user, while another forensic VM (with the Parrot OS) is in charge of monitoring the user's actions. These VMs are isolated from the Internet during the analysis phase, although communication in the isolated virtual network is possible. The Internet network is enabled for both during the collection phase involving the synchronization with the Garmin cloud service, in order to monitor the network traffic between the device and the provided cloud service. After this, both VMs are isolated again.

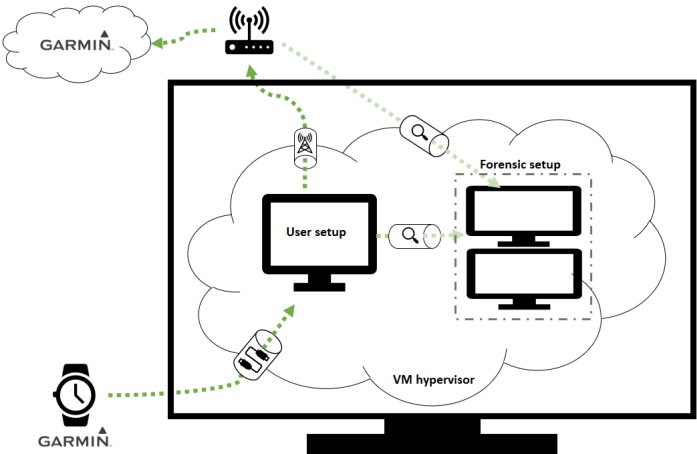

**Figure 1.** Virtual environment architecture.

The principal tools employed in the Parrot OS are:

- Wireshark. A network-monitoring tool. Depending on the network card configuration, this tool allows us to monitor the input and output data of the machine; it can also monitor the whole network in which it participates as a client.
- The Aircrack suite. Composed of tools to evaluate the WiFi network security. In particular, Airodump-ng and Aireplay-ng are used to decrypt network packets related to device synchronization.

The forensic Windows VM incorporated into the virtual laboratory for memory analysis purposed is isolated from the Internet during the analysis phase. The following memory analysis tools are configured in this VM:

- FTK Imager. A device memory dump tool, used before and after synchronization of the device with the cloud.
- Volatility. A utility framework to carry out forensic analysis of the obtained dumps. It is necessary to have Python (version > 2.7) installed previously on the system. Then, using the command line, it is executed from the script directory previously downloaded from GitHub [26].
- Autopsy. A tool for conducting forensic analysis, collecting and documenting every piece of evidence obtained for the subsequent report.

## 4. Data Investigation

A practical use case with a real sport device was analysed by following a classic methodology for forensic analysis. In particular, a data investigation process was performed for the forensic analysis over a Garmin Forerunner 920XT device. It was based on the UNE 71506:2013 [16] and UNE-EN ISO/IEC 27037:2016 [17] standards.

To achieve this, the emulation of a digital forensic analysis process was performed considering data acquisition, evidence documentation and analysis, and network analysis

for interactions. This study goes deeper by analysing possible vulnerabilities within the interactions between this Garmin device and its cloud services. The data investigation process will provide several security guidelines and recommendations to preserve data privacy.

*4.1. Data Acquisition*

Once the virtual environment was designed and deployed, the user's personal real data was collected. In this use case, we emulated a user carrying the smartwatch all day, monitoring their activity, steps, and so on. Then, the user performed sports activities using a heart rate band paired with the smartwatch and the GPS activated. After this, the device was utilized for the digital forensic procedure. A Garmin Forerunner 920XT was used for this research work. This device includes GPS, an altimeter, a compass, an accelerometer, and sleep and step sensors, among others. In addition, it can be paired with heart rate monitors, power meters, and pedometers, enriching the user's records for different sports activities.

Furthermore, a Polar H10 heart rate sensor was paired with the smartwatch, which was synchronized using the ANT+ protocol. The smartwatch also supports pairing via Bluetooth with a smartphone (which can transfer applications, photos, and so on) or WiFi, supporting direct synchronization with the Garmin cloud service, thus bypassing the smartphone as an access point.

According to the data collection process, it is possible to detect communication packets sent from the smartwatch to the Garmin cloud service with the RFMON (radio frequency monitor) mode enabled, at a technical level. For successful decryption, the packets were captured with Airodump by means of a handshake process. Figure 2 depicts this communication between client and server, with the associated network packets encrypted by the WiFi password.

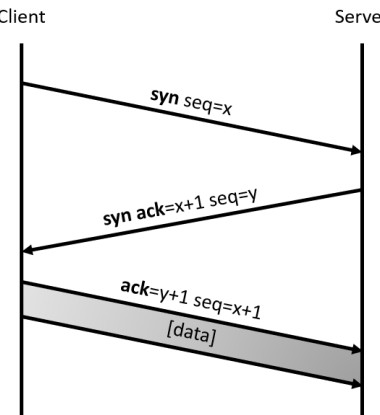

**Figure 2.** *Handshake* diagram.

To obtain these network packets, the client must be manually disconnected from the network (and connected again to it). As an alternative, a de-authentication process can be forced. This latter option was chosen, by means of `aireplay-ng`, in order to successfully capture packets. The decryption was performed in Wireshark, as previously detailed in [27,28].

The main purpose of this step is to collect the network packets generated in the WiFi synchronization process. For this phase, it is necessary to follow the steps detailed in Appendix A.1. When the virtual environment is ready, it is necessary to monitor the network traffic (using airodump-ng) in order to obtain a valid handshake, executing a de-authentication attack in a controlled way with aireplay-ng, such that the same device tries to connect again and the necessary packets for a valid handshake can be collected. It is essential that the evidence is isolated, as the device tries to carry out a previous synchronization via Bluetooth. Appendix A.2 details the data acquisition process employed.

Once the data transfer is completed, the virtual network is disconnected from the WiFi network.

### 4.2. Evidence Documentation

Once the data are collected, before synchronization of the device with the cloud, FTK Imager is used to conduct an image dump in the raw format. All gathered digital evidence from personal data is documented for subsequent forensic analysis. New evidence is added, the physical disk of the device is selected, and the raw format is chosen with the file destination. After a few minutes, a summary of the extraction is returned, as shown in Figure 3. This summary provides information about the evidence (previously entered manually), drive information, and hash verification. This evidence is then included in the opened case with Autopsy, a graphical interface solution which can be used to analyse the evidence with tools such as keyword searches or event timelines, thus facilitating the subsequent forensic analysis in the evidence analysis phase.

```
Created By AccessData® FTK® Imager 4.5.0.3

Case Information:
Acquired using: ADI4.5.0.3
Case Number: 001
Evidence Number: 001
Unique description: garmin_previous_sincronization
Examiner: Pablo Donaire
Notes: Pre-synchronization memory dump

-------------------------------------------------------------

Information for D:\garmin_previous_sincronization:

Physical Evidentiary Item (Source) Information:
[Device Info]
 Source Type: Physical
[Drive Geometry]
 Cylinders: 1
 Tracks per Cylinder: 255
 Sectors per Track: 63
 Bytes per Sector: 512
 Sector Count: 22.477
[Physical Drive Information]
 Drive Model: Garmin FR920 FLASH USB Device
 Drive Interface Type: USB
 Removable drive: True
 Source data size: 10 MB
 Sector count:    22477
[Computed Hashes]
 MD5 checksum:    b00cbc034ebced9fb5575e756f3fe904
 SHA1 checksum:   a1ade640cb718c42d3065dd15ebccdd3fcf8917e

Image Information:
 Acquisition started:   Sun May  1 17:58:07
 Acquisition finished:  Sun May  1 18:04:46
 Segment list:
  D:\garmin_previous_sincronization.001

Image Verification Results:
 Verification started:  Sun May  1 18:04:46
 Verification finished: Sun May  1 18:04:47
 MD5 checksum:    b00cbc034ebced9fb5575e756f3fe904 : verified
 SHA1 checksum:   a1ade640cb718c42d3065dd15ebccdd3fcf8917e : verified
```

**Figure 3.** Extraction summary of a device memory dump (before synchronizing with the cloud).

Regarding the memory dump collection, after the synchronization process with the cloud, the general procedure is the same as that for the previous dump. The summary of the new extraction is detailed in Figure 4, where the information is almost the same as before, except for the computed hashes varying due to information having been modified since the last dump.

```
Created By AccessData® FTK® Imager 4.5.0.3

Case Information:
Acquired using: ADI4.5.0.3
Case Number: 001
Evidence Number: 003
Unique description: garmin_post_sincronization
Examiner: Pablo Donaire
Notes: Post-synchronization memory dump

-------------------------------------------------------------

Information for D:\garmin_posterior_sincronization:

Physical Evidentiary Item (Source) Information:
[Device Info]
 Source Type: Physical
[Drive Geometry]
 Cylinders: 1
 Tracks per Cylinder: 255
 Sectors per Track: 63
 Bytes per Sector: 512
 Sector Count: 22.477
[Physical Drive Information]
 Drive Model: Garmin FR920 FLASH USB Device
 Drive Interface Type: USB
 Removable drive: True
 Source data size: 10 MB
 Sector count:    22477
[Computed Hashes]
 MD5 checksum:     7b9645c7897d94971f07c246e8aa1979
 SHA1 checksum:    64acaee5ac57bff5f094eae9bc0a165b45816b60

Image Information:
 Acquisition started:   Sun May  1 18:38:05
 Acquisition finished:  Sun May  1 18:44:43
 Segment list:
  D:\garmin_posterior_sincronization.001

Image Verification Results:
 Verification started:  Sun May  1 18:44:43
 Verification finished: Sun May  1 18:44:44
 MD5 checksum:     7b9645c7897d94971f07c246e8aa1979 : verified
 SHA1 checksum:    64acaee5ac57bff5f094eae9bc0a165b45816b60 : verified
```

**Figure 4.** Extraction summary of a device memory dump (after synchronizing with the cloud).

In order to collect data from the Garmin cloud application, it is necessary to check whether the desktop application supports Garmin Express, which offers additional information about the synchronization process. To synchronize data, it is necessary to have a charging cable. As mentioned in [18], this is one of the main drawbacks regarding the forensic analysis of these devices: in most cases, it is necessary to use an exclusive cable for each watch.

In our case, two kinds of dumps are collected as evidence for their subsequent analysis:

- System monitoring to check whether new files and/or memory registers have been generated. As data dumps received with Process Monitor can be quite complex to analyse, we use Noriben [29], a Python script that organizes this information in a much more understandable way, leaving two files as a result of monitoring: A .csv file and a .pml file. These can be opened with Process Monitor for analysis without filtering via the script.
- Network monitoring with Wireshark, which captures every communication between the application and the cloud. Wireshark monitoring is executed during Garmin Express synchronization, in the same way as before. Later during the analysis, the previous monitoring is used to search for those communications carried out via the application.

### 4.3. Evidence Analysis

Prior to synchronization, in an environment without networks where the device could connect (WiFi or Bluetooth networks), the information dump is conducted using FTK

Imager (as explained in Section 4.1). This section covers the analysis of the information previously collected and documented with Autopsy.

After opening the memory dump in Autopsy, the data are organized as follows:

- GARMIN/ACTIVITY: Directory with every activity stored in the device previously recorded.
- GARMIN/APPS: Directory where information concerning installed widgets on the device is stored.
- GARMIN/COURSES: Directory with tracks pre-loaded by the user.
- GARMIN/EVNTLOGS: Directory with the system logs.
- GARMIN/GOALS: Directory with the daily goals set by the user (e.g., number of steps by day).
- GARMIN/LOCATION: Directory with the waypoints stored by the user.
- GARMIN/MLTSPORT: Directory with multi-sport activities (i.e., triathlon).
- GARMIN/MONITOR: Directory with the compilation of daily activities. This device does not monitor them with GPS but with a built-in pedometer. Therefore, it only records the distance and calories burned during segments of movement and rest. Calculation of both is carried out using mathematical methods based on parameters that are indicated at the beginning by the user.
- GARMIN/NEWFILES: File reception directory from the cloud to the device. This folder should be emptied after a certain time, at which point any .fit file should be deleted by the device and stored in memory.
- GARMIN/RECORDS: Contains a single file RECORDS.FIT. In this file, the personal records set by the user in different activities are collected (e.g., best time running 1 km, 1 mile, half marathon).
- GARMIN/REMOTESW: Despite this directory being empty, Autopsy automatically conducts a carving process, finding different BIN and RGN (Garmin region file) files, which are responsible for the different system updates.
- GARMIN/SCHEDULE: Contains a SCHEDULE.FIT file with internal system data.
- GARMIN/SETTINGS: Contains a SETTINGS.FIT file with all user settings (e.g., age, weight, height, maximum heart rate, rhythm zones).
- GARMIN/SPORTS: Contains a specific file for each type of sport, in which it is possible to configure the specific training settings for that activity (e.g., heart rate, power).
- GARMIN/TEMPFIT: Directory where ongoing activities are stored. This folder contains files when some activity is in progress. Considering the way that this device proceeds when it deletes information (logical deletion), it is to be assumed that it moves files instead of removing them.
- GARMIN/TEXT: Directory with .ln2 files, which include labels that are displayed. Each of them contains every label for a specific language.
- GARMIN/TOTALS: Contains a TOTALS.FIT file with the sum of activities, distance, and time of each modality.
- GARMIN/WIFI: Contains an encrypted OUT.BIN binary file, in which the WiFi information is stored, in a similar way as shown in Figure 5.

**Figure 5.** OUT.BIN File encrypted.

- GARMIN/WORKOUTS: Directory with planned activities by the user (i.e., workouts).

Files found in every directory include FIT and BIN extension files. .FIT files can be opened using public tools, such as Fit File Viewer [30], while the binary files are encrypted and were impossible to open. These .FIT files that correspond to activities are quite complex

to read; however, they can be displayed as maps using tools such as GPS Visualizer [31], allowing the workout route to be displayed in a proper way.

Prior to opening the captured frame in Wireshark, we opened the watch configuration to acquire the MAC address (`10:c6:fc:d4:93:88`). With this, filtering was performed, specifically looking for `wlan.sa == 10:c6:fc:d4:93:88 || wlan.da == 10:c6:fc:d4:93:88`, filtering every packet for which the source or destination MAC is the smartwatch (see Figure 6).

**Figure 6.** First smartwatch communication.

We highlight the following group of packets:

- Group 1: As mentioned regarding the environment configuration in Appendix A.1, when a device accesses the network, it must agree with the AP.
- Group 2: Two ARP communications on the network, reporting the IP assigned to the smartwatch to keep the address mapping updated.
- Group 3: Requests for a resolution to Movistar DNS of a garmin.com sub-domain, receiving a response in the following packets. It is appreciated that the infrastructure set up is a Cloudflare CDN server. As port scanning without authorization is illegal, it is not possible to check what services are offered through the mentioned IP.

Next, the communication between the smartwatch client and the Garmin server begins. In this case, as what is interesting is the data communicated between the two, a second screening was carried out: (`wlan.sa == 10:c6:fc:d4:93:88 || wlan.da == 10:c6:fc:d4:93:88`) and (`http.request or tls.handshake.type eq 1`), filtering every packet for which the MAC source or destination is the smartwatch and ensures that the packets belong to an HTTP or HTTPS request). A total of 19 results were obtained (see Figure 7).

Further examining the TCP flows above, some interesting facts can be noted. In packet number 5835, when reconstructing the packet chain (using `right click > Follow > TCP flow`), two interesting facts stood out:

- There is a `gcs-return-response-length` header. The `gcs-` prefix indicates that the service is working with the Google Cloud Storage platform (see Figure 8). Although this service is behind Cloudflare, there are free websites such as ImmuniWeb that can offer this information [32].
- An XML file is sent with each communication, with the information that the device should exchange with the cloud (see Figure 8).

| No. | Source | Destination | Protocol | Length | Server Name | Info |
|---|---|---|---|---|---|---|
| 5257 | 192.168. | 104.19. | HTTP | 255 | | POST /OBN/OBNServlet HTTP/1.1 |
| 5464 | 192.168. | 104.19. | HTTP | 104 | | POST /OBN/OBNServlet HTTP/1.1 |
| 5835 | 192.168. | 104.19. | HTTP | 1266 | | POST /OBN/OBNServlet HTTP/1.1 |
| 5903 | 192.168. | 104.19. | HTTP | 304 | | POST /OBN/OBNServlet HTTP/1.1 |
| 5970 | 192.168. | 104.19. | HTTP | 389 | | POST /OBN/OBNServlet HTTP/1.1 |
| 6158 | 192.168. | 104.19. | HTTP | 410 | | POST /OBN/OBNServlet HTTP/1.1 |
| 6543 | 192.168. | 104.19. | HTTP | 272 | | POST /OBN/OBNServlet HTTP/1.1 |
| 6630 | 192.168. | 104.19. | HTTP | 1167 | | POST /OBN/OBNServlet HTTP/1.1 |
| 6726 | 192.168. | 104.19. | HTTP | 968 | | POST /OBN/OBNServlet HTTP/1.1 |
| 6793 | 192.168. | 104.19. | HTTP | 1274 | | POST /OBN/OBNServlet HTTP/1.1 |
| 6845 | 192.168. | 104.19. | HTTP | 1312 | | POST /OBN/OBNServlet HTTP/1.1 |
| 6891 | 192.168. | 104.19. | HTTP | 752 | | POST /OBN/OBNServlet HTTP/1.1 |
| 6963 | 192.168. | 104.19. | HTTP | 654 | | POST /OBN/OBNServlet HTTP/1.1 |
| 7031 | 192.168. | 104.19. | HTTP | 667 | | POST /OBN/OBNServlet HTTP/1.1 |
| 7119 | 192.168. | 104.19. | HTTP | 655 | | POST /OBN/OBNServlet HTTP/1.1 |
| 7213 | 192.168. | 104.19. | HTTP | 834 | | POST /OBN/OBNServlet HTTP/1.1 |
| 7279 | 192.168. | 104.19. | HTTP | 769 | | POST /OBN/OBNServlet HTTP/1.1 |
| 7319 | 192.168. | 104.19. | HTTP | 781 | | POST /OBN/OBNServlet HTTP/1.1 |
| 7497 | 192.168. | 104.19. | HTTP | 270 | | POST /OBN/OBNServlet HTTP/1.1 |

**Figure 7.** Data exchange between the smartwatch and the cloud.

**Figure 8.** Header included in the communication regarding Google Cloud Storage.

Subsequently, the encrypted data are sent (packets 6630, 6793, and 6845), from which it was only possible to extract the endpoint of the upload activities; thus, presumably, the FIT files with sports activities are sent there, but encrypted under an SSL certificate (see Figure 9).

**Figure 9.** Encrypted file with activity.

Next, the memory dump after synchronization was analysed. As the data structure is the same as that seen in the analysis prior to the synchronization, this section only provides a comparison between both, looking for differences that exist after WiFi synchronization:

- ACTIVITY directory: There are no changes after activity synchronization, so there is no automatic deletion of activities.
- APPS, COURSES, GOALS, LOCATION, TEMPFIT, TEXT, WORKOUTS directories: Without changes. This is due to the fact that, in these directories, changes would be made through data transmitted from Garmin Connect, which could give information regarding the time of the last synchronization.
- EVNTLOGS directory: Two files are kept in this directory, removing the oldest one. The file names follow a hexadecimal notation, with 00000000.TXT being the first log file and, for each synchronization, the file name is increased by one. The first lines of the file contain the information presented in Figure 10. Although the Garmin documentation is poor, at a forensic level, the most interesting value would be the

timeDay property, which indicates the time at which the device is turned on. It does not imply that it was turned on for an activity, just that it started working.

- MONITOR directory: In this directory, file deletion is conducted after synchronization. As the synchronization is carried out when the watch is on, a single file is kept, recording the movements at the time of synchronization. Time is the same as that displayed by the watch, which is synchronized on every upload and during GPS synchronization (before an outdoor activity).
- Although there is no file, the carving technique detected files that were removed during synchronization. When it was extracted, it could be seen that there were activities (.FIT files) that had been synchronized.
- RECORDS directory: Counters are updated.
- REMOTESW directory: Using carving, it was verified that the removed file _PO.BIN was updated and removed; however, it was not possible to obtain the content in order to determine which information had been modified.

```
>>>>> EVL_CURR_LOGS <<<<<
verNum:     8/13/2014 (00)
ownerID:    00000008
OwnerName:  TFS
status:     03
pri_thresh: 2
archived:   10
max_arch:   10
num_recs:   20
rec_size:   20
next_evnt:  10
timeDay:    2022/ 5/ 1 16:36:05
cycle_cnt:  316
```

**Figure 10.** Log File from the EVNTLOGS directory.

As previously mentioned, although workouts are not removed automatically, users can remove them manually; however, this deletion is not enough as the files can be recovered through Autopsy (see Figure 11).

In summary, the main differences between the pre- and post-synchronization contexts lie in the deletion of files in directories. However, these files can be recovered through the use of carving techniques. The log files do not provide too much valuable information about the files or communications that have been carried out, but can give some references about the time and date of the last synchronization.

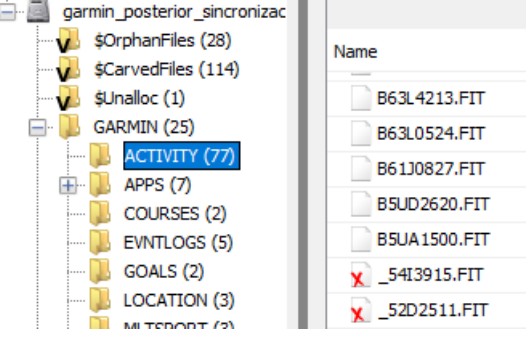

**Figure 11.** Carving into the activities directory.

In addition, our data investigation process also includes analysing possible vulnerabilities during the network interactions among the Garmin device and its associated cloud services, as detailed in the next section.

### 4.4. Network Analysis

After analysing the data evidence obtained from the smartwatch, we analysed the network communications that occurred during the synchronization between the smartwatch and the Garmin cloud.

First of all, we filtered every process that was not of interest, leaving only those related to `express.exe` marked and leaving blank the other processes that were beyond the scope of our investigation (most being internal processes or related to other background applications; see Figure 12).

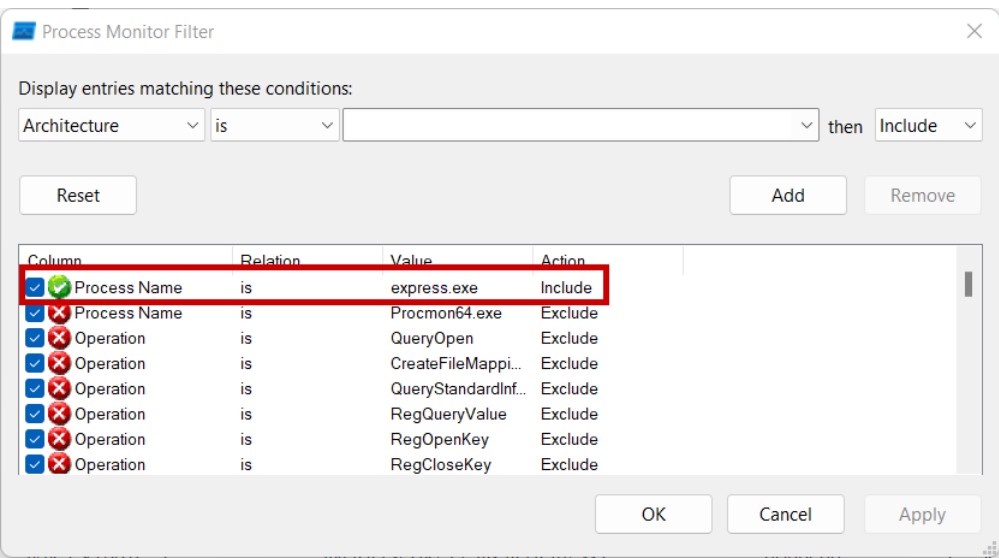

**Figure 12.** Events filtered in process monitoring.

First, we noticed that it starts writing in the `Express.log` and `ExpressDetailed.log` files, where the first is just a condensed version of the latter. When opened, only messages related to the synchronization process were shown, trying to upload `.fit` files and later download updates to the cloud. More valuable information is provided by the latter file (see Figure 13).

```
"Type":3,"DataFormat":2,"ContentType":null},{"Name":"Content-
on/json","Type":3,"DataFormat":2,"ContentType":null},{"Name":"Date","Value":'
t":2,"ContentType":null},{"Name":"Expires","Value":"0","Type":3,"DataFormat":
nseUri" "https://████████.garmin.com/geolocation/whereami/akamai","errorMe
```

**Figure 13.** Log capture revealing Akamai usage.

After modification of the files, the Garmin client starts to communicate with external IPs, sending information (see Figure 14).

Next, a link which refers to Akamai—a widely used CDN provider—was verified. This URL returns only the country where the request was performed.

Next, a URL related to statistical data from Cloudflare was observed, which had already been detected.

Subsequently, multiple encrypted data appeared, from which—although no information was found—it could be seen that a link to the Garmin API appears, where information about the satellite's location is received. This is used by Garmin to allow their watches to pick up a signal faster, improving the user experience [33]. Although it is not possible to ensure which segment refers to the number 28 that appears as a parameter in the URL, it may be possible to create (with time and dedication) a map that allows for locating (with a level of precision not known at this point, as that goes beyond the scope of our study) the point at which the watch is synchronizing. As a possible starting point of this work, say that, at the time of data collection, the watch was synchronized in Madrid, whose post code starts with 28 (the same number as the URL parameter), and could be relying on the

response to the previous request to Akamai (see Figure 13). Finally, the synchronization process is detailed, without more relevant information for the analysis.

| Process Name | PID | Operation | Path |
|---|---|---|---|
| express.exe | 9596 | WriteFile | C:\ProgramData\Garmin\Logs\Express\Express.log |
| express.exe | 9596 | WriteFile | C:\ProgramData\Garmin\Logs\Express\ExpressDetailed.log |
| express.exe | 9596 | WriteFile | C:\ProgramData\Garmin\Logs\Express\Express.log |
| express.exe | 9596 | WriteFile | C:\ProgramData\Garmin\Logs\Express\ExpressDetailed.log |
| express.exe | 9596 | Thread Create | |
| express.exe | 9596 | Thread Create | |
| express.exe | 9596 | TCP Connect | 192.168. :62656 -> 104.18. :443 |
| express.exe | 9596 | Thread Create | |
| express.exe | 9596 | TCP Connect | 192.168. :62657 -> 104.18. :443 |
| express.exe | 9596 | Thread Create | |
| express.exe | 9596 | Thread Create | |
| express.exe | 9596 | Thread Create | |
| express.exe | 9596 | Thread Create | |
| express.exe | 9596 | TCP Send | 192.168. :62656 -> 104.18. :443 |
| express.exe | 9596 | TCP Send | 192.168. :62657 -> 104.18. :443 |
| express.exe | 9596 | TCP TCPCopy | 192.168. :62656 -> 104.18. :443 |
| express.exe | 9596 | TCP Receive | 192.168. :62656 -> 104.18. :443 |
| express.exe | 9596 | TCP Receive | 192.168. :62656 -> 104.18. :443 |
| express.exe | 9596 | TCP TCPCopy | 192.168. :62656 -> 104.18. :443 |
| express.exe | 9596 | TCP Receive | 192.168. :62656 -> 104.18. :443 |
| express.exe | 9596 | TCP TCPCopy | 192.168. :62657 -> 104.18. :443 |
| express.exe | 9596 | TCP Receive | 192.168. :62657 -> 104.18. :443 |
| express.exe | 9596 | TCP Receive | 192.168. :62657 -> 104.18. :443 |
| express.exe | 9596 | TCP TCPCopy | 192.168. :62657 -> 104.18. :443 |
| express.exe | 9596 | TCP Receive | 192.168. :62657 -> 104.18. :443 |

**Figure 14.** Process Monitor showing the filtered events.

Following the process monitoring analysis, the written files are `device_data_store.xml`, `devices_list.xml` and `preloaded_maps.xml`, which do not provide useful information as they record internal information for the application.

Regarding network communications, there are different destination addresses to Cloudflare CDN through the HTTPS protocol that are reached from different network ports, always grouped in a range starting from 60,000. With the Shodan search engine, we checked that these IP addresses belong to Garmin cloud services.

The entire connection dump was filtered according to the output ports detected by Process Monitor. After investigating the different packets, it can be seen that data are encrypted, preventing information about the activities or user from being obtained (which was foreseeable, as the destination port was always 443, usual for the HTTPS protocol). In any case, it can be assumed that the information sent through this communication includes the different workouts previously recorded by the device.

## 5. Discussion and Recommendations

After performing a forensic analysis focused on the Garmin Forerunner 920XT device, as a practical case of our proposed solution, we confirmed that the sensitive data remain confidential. All data sent during the synchronization process, either by the client application or through a direct connection with the Garmin cloud service, was encrypted.

The tested smartwatch automatically removes information related to daily activities (e.g., steps). However, this information does not pose any risk of violating user privacy, as it does not use GPS. Thus, it is impossible to determine the location of the device. On the other hand, activities that use the GPS sensor (e.g., running, cycling, or open water swimming, among others) are permanently kept in the memory until the user manually removes a specific activity. Furthermore, it would be possible to recover associated data with carving techniques, as long as the information has not been overwritten.

Internally, the smartwatch generates a log file with a consecutive name to the previous one. This allows tracking of the number of times the device has been turned on, the last time it was turned on, or even if the registry has been altered.

These activities are recorded in `.fit` files, which can be easily read by a user without advanced technical skills using public and free tools. These allow for the analysis of such information, or even displaying it on a map. These files carry both the information on

the track followed, as well as the completion date or different biorhythms that have been tracked by sensors (e.g., heart rate, power). This is a security issue as, with simple analysis techniques, a malicious user can discover user patterns or generate heat maps with their most common locations (usually workplaces or homes [34]).

As for more sensitive information, such as pairing with other measurement devices or information related to WiFi signals known by the device, it is stored in encrypted `BIN` files. This information was not possible to read in our case.

Concerning the data stored in the cloud, thanks to the IP and URLs found in the intercepted communications, we discovered that the information is registered on Google Cloud Storage services. The corporate security level configuration of this platform is unknown, but there are already documented cases of data leaks due to misconfigurations [35].

Users can access this information through the Garmin Connect platform, from which they can obtain a correctly structured activity report. However, this information may remain open to the public in case of mismanagement of data privacy by the user. Through advanced use of the Google search engine, it is possible to view the activities of different users, where both the start and end point of each can be clearly observed; for example, a Google search such as `inurl:garmin inurl:connect ''activity''` can be carried out to see certain activities which, in turn, lead to user profiles. This search can be extended to other social web applications, such as Strava; however, this is beyond the scope of the current study.

To preserve the information that users share with sport devices, it would be advisable to follow the security guidelines offered below, in order to guarantee the privacy of user data:

- Data synchronization via WiFi should only be performed on confidence networks with minimum security levels [36], thus avoiding possible personal data leaks.
- Once sport activities are synchronized, they must be removed from the device. Although deletion is not achieved with wiping, the information may be totally or partially overwritten by new activities, making it impossible to recover. Therefore, the risk of recovering that information is lessened over time.
- It is not recommended to start a sport activity close to one's place of residence or work, as it can allow for the detection of patterns or usual locations.
- In case total confidentiality is desired by the smartwatch user, it is advisable to delete the associated data after each sport activity. In this way, the smartwatch can synchronize with a GPS satellite location detection and forecast system for several activities to aid synchronization with subsequent activities, thus improving its quality of service. However, the principal drawback of this approach is that data is saved in a specific file, which covers a radius of action that is too large to accurately locate a person. By following this recommendation, this file must be deleted from the smartwatch.

The principal limitation of this study was the lack of public documentation provided by Garmin regarding its API interface. In some cases, it was essential to study in-depth other works from the forensic community. In addition, a forensic analysis of the communication carried out between the cloud services in charge of storing Garmin's activities and the user's device is offered, proving that the security levels are required to preserve sensitive data. A very promising work for the forensic analysis of wearable devices from the logical and physical perspective was proposed in [24]. Specifically, a set of Samsung and Apple smartwatches, and the Vivosport smartband. According to this, no other specific studies deal with Garmin smartwatches (in our case, the Garmin Forerunner 920XT) in the context of forensic analysis for formative purposes and without manipulating the device during the forensic process. An exhaustive evidence and network analysis, as well as cloud interactions, were performed with an emulated virtual environment considering user data privacy for the different types of analyses. We also provided a set of recommendations and guidelines for these kind of devices.

## 6. Conclusions

The forensic analysis of smartwatches is increasingly becoming very relevant in the daily lives of their users. The amount of information that these devices can record, together

with ignorance about user data protection, poses risks regarding the integrity of this information if it falls into the wrong hands. Data protection in these devices may also be essential for the resolution of cases. Fortunately, devices such as the Garmin smartwatch analysed in this work are normally very concerned with user privacy, encrypting all data that are transferred to the Internet and avoiding any information leakage. This confidentiality would be complete if data were kept encrypted within the smartwatch; however, this may affect the user experience in terms of performance. Additionally, users must be aware of data privacy. Therefore, some security recommendations were provided.

Regarding further works, as the synchronization of the analysed sport device with the cloud is also possible via the Bluetooth protocol, it would be of great interest to assess such a connection in forensic analysis. However, VMs that simulate Android-based devices are not yet ready to emulate this functionality. When this issue is resolved, it would be interesting to investigate the above-mentioned connection in depth. Likewise, similar tests could be carried out in an iOS simulation environment, checking whether confidentiality is maintained in either context.

**Author Contributions:** Conceptualization, P.D.-C. and L.T.; methodology, A.R.-G., L.T. and R.P.-V.; software, P.D.-C.; validation, P.D.-C., L.T. and R.P.-V.; formal analysis, L.T.; investigation, P.D.-C., A.R.-G., L.T. and R.P.-V.; resources, A.R.-G. and R.P.-V.; data curation, P.D.-C., L.T. and R.P.-V.; writing—original draft preparation, P.D.-C. and A.R.-G.; writing—review and editing, P.D.-C., A.R.-G., L.T. and R.P.-V.; visualization, P.D.-C.; supervision, A.R.-G., L.T. and R.P.-V.; project administration, A.R.-G.; funding acquisition, A.R.-G. and R.P.-V. All authors have read and agreed to the published version of the manuscript.

**Funding:** This work was funded by Universidad Nacional de Educación a Distancia (UNED), under the SUMA-CITeL research project (grant number 096-043077).

**Data Availability Statement:** Not applicable.

**Acknowledgments:** Authors would like to acknowledge the support of the I4Labs UNED research group, the CiberGID UNED innovation group with the CiberScratch 2.0 project, and the SUMA-CITeL research project (096-043077) for the 2022–2023 period, as well as the E-Madrid-CM Network of Excellence (S2018/TCS-4307). The authors also acknowledge the support of SNOLA, officially recognized Thematic Network of Excellence (RED2018-102725-T) by the Spanish Ministry of Science, Innovation and Universities.

**Conflicts of Interest:** The authors declare no conflict of interest.

## Abbreviations

The following abbreviations are used in this manuscript:

| | |
|---|---|
| AP | Access point |
| ARP | Address resolution protocol |
| BSSID | Basic service set identifier |
| CDN | Content delivery network |
| GPS | Global positioning system |
| HTTP | Hypertext transfer protocol |
| HTTPS | HyperText transfer protocol secure |
| IP | Internet protocol |
| iOS | Intelligent operative system |
| MAC | Media access control |
| OS | Operating system |
| RFMON | Radio frequency monitor |
| SSL | Secure sockets layer |
| TCP | Transmission control protocol |
| URL | Uniform resource locator |
| VM | Virtual machine |
| WPA2 | WiFi protected access 2 |
| XML | Extensible markup language |

**Appendix A**

*Appendix A.1. Network Card Configuration*

The network card configuration to monitor the traffic of the defined environment is shown. Wireshark is employed for this purpose. The network card must support the RFMON or Monitor mode. In this case, a TP-Link TL-WN722N is used with this mode enabled. The steps to change this configuration from the VM have been previously detailed [27].

As we are working on a VM, it is mandatory to enable a USB connection. To do this, in the VirtualBox Administrator, the VM is selected and `Configuration > USB` is clicked. Then, the device is added by clicking the second icon on the right.

Then, it should be possible to see the available network cards from a VM terminal, and the mode is enabled with the command `iw dev` (see Figure A1).

```
┌─[osboxes@osboxes]─[~]
└──╼ $iw dev
phy#0
        Interface wlx54e6fc872407
                ifindex 3
                wdev 0x1
                addr ae:8c:▓▓▓▓▓▓
                type managed
                txpower 20.00 dBm
```

**Figure A1.** Available interfaces.

With the interface name and observing that it is in the managed mode (only packets that have the host MAC address as the destination are captured), we proceed to deactivate the interface, enable it in Monitor mode, and activate it again, as shown in Figure A2).

```
┌─[root@osboxes]─[/home/osboxes]
└──╼ #ip link set wlx54e6fc872407 down
┌─[root@osboxes]─[/home/osboxes]
└──╼ #iw wlx54e6fc872407 set monitor control
┌─[root@osboxes]─[/home/osboxes]
└──╼ #ip link set wlx54e6fc872407 up
```

**Figure A2.** Monitor mode on.

After the `iw dev` command is executed again, the interface is now in Monitor mode (see Figure A3). This mode is maintained when the network card is connected and the user does not log out; otherwise, these steps must be repeated to enable this mode again.

```
┌─[root@osboxes]─[/home/osboxes]
└──╼ #iw dev
phy#0
        Interface wlx54e6fc872407
                ifindex 3
                wdev 0x1
                addr fe:3e:46:15:a4:b5
                type monitor
                channel 1 (2412 MHz), width: 20 MHz (no HT), center1: 2412 MHz
                txpower 20.00 dBm
```

**Figure A3.** Monitor model interface.

*Appendix A.2. Data Acquisition Process*

The data acquisition process for the proposed case study is now detailed. First, the command `airodump-ng <INTERFACE_NAME>` is executed with the RFMON network mode enabled. The list of wireless networks visible to the card is shown, as shown in Figure A4. The most relevant indicators for our purposes are BSSID and transmission channel.

```
CH 11 ][ Elapsed: 6 s ][ 2022-04-30 13:58

BSSID              PWR  Beacons   #Data, #/s  CH   MB  ENC  CIPHER  AUTH ESSID

70:97:41          -88      1        0    0   11  130  WPA2  CCMP   PSK
10:50:             -87      2        0    0   11  130  WPA2  CCMP   PSK
18:E8:             -87      2        0    0   11  130  WPA2  CCMP   PSK
08:6A:0A:BB:BE:4A  -66      5        5    0   11  130  WPA2  CCMP   PSK  MOVISTAR_BE49
E6:AB:             -67      7        2    0    6  130  WPA2  CCMP   PSK
E4:AB:             -67      7        0    0    6  130  WPA2  CCMP   PSK
FA:8F:             -68      9        0    0    7   65  OPN
E8:D8:             -68      3        0    0    1   65  WPA2  CCMP   PSK
CC:D4:             -71      2       61    0    1  130  WPA2  CCMP   PSK
00:31:             -74      6        0    0    6  130  WPA2  CCMP   PSK
E0:19:             -74      8        6    0    7  130  WPA2  CCMP   PSK
78:DD:             -76      2        0    0   11  130  WPA2  CCMP   PSK
50:78:             -77      4        0    0    6  195  WPA2  CCMP   PSK
FA:8F:             -78      5        0    0    6   65  OPN
60:8D:             -80      0        3    0    1   -1  WPA
30:CC:             -85      2        0    0    4  130  WPA2  CCMP   PSK
90:9A:             -87      3        0    0    1  540  WPA2  CCMP   PSK

BSSID              STATION              PWR   Rate    Lost    Frames  Notes  Probes

08:6A:0A:BB:BE:4A  DC:4F:22:D6:2C:66    -60   0 - 6     0       1
08:6A:0A:BB:BE:4A  DC:4F:22:D6:03:EB    -66   0 - 6     0       1
CC:D4:             9C:28:               -84  24e- 1e   714     64
E0:19:             70:EE:               -1   1e- 0      0       4
E0:19:             E4:F0:               -61   0 -24e    0       1
60:8D:             80:C5:               -60  6e- 6e     0       3
```

**Figure A4.** List of BSSIDs.

By using these indicators, network packets are captured with the command `airodump-ng -c <CHANNEL> -bssid <BSSID> -w <DUMP_FILES_NAMES> <INTERFACE_NAME>`, where:

- <CHANNEL> is the wireless network channel being captured;
- <BSSID> is the address from which packets are obtained, filtering noise from the rest of the APs transmitted on the previous channel;
- <DUMP_FILES_NAMES> are the files generated after the capture process;
- <INTERFACE_NAME> is the name of the network interface used to monitor the wireless network.

Figure A5 shows the result of executing the previous command, where we see a summary of our network monitoring and, at the bottom, a list of devices connected to the same network, showing their MAC address, data rate, and so on.

```
CH 11 ][ Elapsed: 24 s ][ 2022-04-30 16:56

BSSID              PWR RXQ  Beacons    #Data, #/s  CH   MB   ENC  CIPHER  AUTH ESSID

08:6A:0A:BB:BE:4A  -63  90     216       197    0  11  130   WPA2 CCMP    PSK  MOVISTAR_BE49

BSSID              STATION            PWR   Rate    Lost   Frames  Notes  Probes

08:6A:0A:BB:BE:4A  70:EE:50:0D:2D:24  -57  24e-12e    0     18
08:6A:0A:BB:BE:4A  DC:4F:22:D6:2C:66  -58   6e- 6     2    116
08:6A:0A:BB:BE:4A  DC:4F:22:D6:03:EB  -65   6e- 6     4     85
08:6A:0A:BB:BE:4A  3C:5C:C4:C7:A7:83  -67   1e- 1e    0      3
08:6A:0A:BB:BE:4A  C8:C2:FA:E0:74:28  -71   1e- 1    19      7
```

**Figure A5.** Network packet capture.

In the same way, a new terminal will be opened to force the client's de-authentication. With this step, we force the connection to the network in a controlled way, thus obtaining the necessary packets to carry out a successful handshake (Figure A6). The specific command used to perform this action is `aireplay-ng -0 1 -a <BSSID> -c <CLIENT_MAC> <NETWORK_INTERFACE>`, where the `-0 1` flag enables the de-authentication mode in aireplay-ng and the number of attempts sent (can be multiple) and:

- <BSSID> is the MAC address of the access point transmitting de-authentication packets (in this case, the same are used to monitor the wireless network);
- <MAC_CLIENT> is the client de-authentication, obtained from our last `airodump-ng` command, where we see a list of connected devices.

```
┌─[root@osboxes]─[~]
└─→ #aireplay-ng -0 1 -a 08:6A:  :BB:  :  -c 4C:EB:  :B9:  :  wlx54e6fc872407
17:00:44  Waiting for beacon frame (BSSID: 08:6A:  :BB:  :  ) on channel 11
17:00:45  Sending 64 directed DeAuth (code 7). STMAC: [4C:EB:  :B9:  :  ] [15|56 ACKs]
```

**Figure A6.** Forced client de-authentication.

Then, the capture registers the four messages needed to generate the handshake protocol, as shown in Figure A7. After synchronization is completed, the monitoring

process with `airodump-ng` can be stopped. It generates five files, but only the `*.cap` file is analysed with Wireshark, analysing the content of packets communicated through the network.

```
CH 11 ][ Elapsed: 2 mins ][ 2022-04-30 16:57 ][ WPA handshake: 08:6A:▮▮ ▮▮ ▮▮ ▮▮

BSSID             PWR RXQ  Beacons    #Data, #/s  CH  MB   ENC CIPHER AUTH ESSID

08:6A:0A:BB:BE:4A -68 100     1118      910   12  11 130   WPA2 CCMP  PSK  MOVISTAR_BE49

BSSID             STATION          PWR   Rate    Lost   Frames  Notes  Probes

08:6A:▮▮ ▮▮ ▮▮ ▮▮  4C:EB:▮▮ ▮▮ ▮▮ ▮▮  -26   1e- 1e    26     140  PMKID
08:6A:▮▮ ▮▮ ▮▮ ▮▮  70:EE:▮▮ ▮▮ ▮▮ ▮▮  -54  24e-12e     2     115
08:6A:▮▮ ▮▮ ▮▮ ▮▮  DC:4F:▮▮ ▮▮ ▮▮ ▮▮  -57  24e- 6     12     544
08:6A:▮▮ ▮▮ ▮▮ ▮▮  3C:5C:▮▮ ▮▮ ▮▮ ▮▮  -65   1e- 1      0      13
08:6A:▮▮ ▮▮ ▮▮ ▮▮  DC:4F:▮▮ ▮▮ ▮▮ ▮▮  -66  24e- 6    445     435
08:6A:▮▮ ▮▮ ▮▮ ▮▮  C8:C2:▮▮ ▮▮ ▮▮ ▮▮  -71   1e- 1      0      28
```

**Figure A7.** Completed handshake process.

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
