# Peer review of "Forensic Analysis Laboratory for Sport Devices: A Practical Use Case"

_electronics, doi:10.3390/electronics12122710_

Round 1
Reviewer 1 Report
[Introduction]
- Provide greater context of the research problem (at the moment you have only indicated Spain)
- Please reformulate the study contributions, in particular I think that the last one should be reconsidered.
[Related Work]
- Please allocate your research among other studies, in particular please indicate the research gap. In other words, please clearly indicate what is missing in current body of knowledge in the research field of your interest.
[Methodology]
- Please remove the Microsoft Windows trademarks signs. It is not appropriate scientific practice to use (or even promote) commercial organizations. On the other hand, what are the differences of using MS Windows instead of Linux? Your subject of study does not concern desktop operating systems, so this kind of details are unnecessary to discuss.
- Figure 2 is unnecessary and brings nothing to your paper.
- Figure 3 is unnecessary. However, if you want to describe in details your environment, which is actually a good idea, the best place is an appendix. I would even recommend you to do that, since your study will gain on the reproductivity, and thus would be seen as more reliable.
[A Practical Case Study] - please rename this section. In my humble opinion, “Data investigation” sounds better, however it is still up to you.
- Some figures are truly unnecessary, for instance Figure 11, Figure 12, Figure 13, Figure 14 (especially) hardly contribute to the study.
- Considering my proposition given above (“Data investigation”), the goal of the investigation somehow is missing. What are you looking for?
[Discussion and Recommendations]
- I don’t fully understand your last recommendation (“The smartwatch synchronizes with a GPS satellite”). Could you consider reformulating this paragraph and making it more understandable?
- I would also suggest move Figure 27 from this section to another one.
- Moreover, in Discussion section, please also provide: study limitations, and implications for both theory and practice.
[Conclusions and Further Works] – please rename this section. “Conclusions” will be enough.
Frankly speaking, I want to thank you for reading such valuable study. Considering my perspective, I was comprehensibly informed. I think your study is interesting and timely.
Author Response
Comment/Suggestion:
[Introduction]
- Provide greater context of the research problem (at the moment you have only indicated Spain)
- Please reformulate the study contributions, in particular I think that the last one should be reconsidered.
Answer:
The manuscript has been revised to include a greater context of the research problem, not only in Spain. This way, additional references have been added with additional justifications. Smartwatches and fitness devices are the most popular due to their great mobility and connectivity capabilities. They are continuously connected to users’ mobile device with a large variety of sensors, and they offer notifications, alerts, recommendations, etc. This way, there has been a big increase in the number of physical activities carried out by people. The three contributions of the study have also been reformulated, as indicated by reviewer.
All changes included in the new version of the manuscript are marked in blue color.
Comment/Suggestion:
[Related Work]
- Please allocate your research among other studies, in particular please indicate the research gap. In other words, please clearly indicate what is missing in current body of knowledge in the research field of your interest.
Answer:
The related work of the manuscript has been improved to allocate our research better among other studies, as well as some minor improvements. We consider the protection of user data obtained from such sport devices in this work.
From our knowledge, there is currently not any public research for the Garmin Forerunner 920XT device in the context of forensic analysis. This smartwatch is widely employed by many athletes. The study goes deeper by examining and analyzing possible vulnerabilities within the interaction among this device and the associated cloud services of Garmin.
New additional references have also been added in the new version of the manuscript. The gaps encountered have also emphasized in this section, so increasing the interest of the present study.
All changes included in the new version of the manuscript are marked in blue color.
Comment/Suggestion:
[Methodology]
- Please remove the Microsoft Windows trademarks signs. It is not appropriate scientific practice to use (or even promote) commercial organizations. On the other hand, what are the differences of using MS Windows instead of Linux? Your subject of study does not concern desktop operating systems, so this kind of details are unnecessary to discuss.
- Figure 2 is unnecessary and brings nothing to your paper.
- Figure 3 is unnecessary. However, if you want to describe in details your environment, which is actually a good idea, the best place is an appendix. I would even recommend you to do that, since your study will gain on the reproductivity, and thus would be seen as more reliable.
Answer:
Microsoft Windows trademarks signs have been removed in the Figure 1. Additionally, Figure 2 has been removed, as requested. In addition to this, a new appendix has been created to describe specific details of our environment, as recommended. The text has also been revised and clarified to tackle the proposed improvements. Even, a new footnote has been included in the new version of the manuscript.
All changes included in the new version of the manuscript are marked in blue color.
Comment/Suggestion:
[A Practical Case Study] - please rename this section. In my humble opinion, “Data investigation” sounds better, however it is still up to you.
- Some figures are truly unnecessary, for instance Figure 11, Figure 12, Figure 13, Figure 14 (especially) hardly contribute to the study.
- Considering my proposition given above (“Data investigation”), the goal of the investigation somehow is missing. What are you looking for?
Answer:
The name of this section has been renamed, as indicated. Additionally, these figures have been removed, and others have been moved to a new appendix section. This section has also been revised to emphasize the principal goals of this research (according to the reformulated contributions).
The initial part of the section has been revised to clarify the goal of the research, as well as the focus of the work over Garmin smartwatches. Specifically, the emulation of a digital forensic analysis process is performed by using the designed virtual laboratory. The study goes deeper by analyzing possible vulnerabilities within the interactions among this Garmin device and its cloud services. From the data investigation process, several security guidelines and recommendations will be given to preserve data privacy.
All changes included in the new version of the manuscript are marked in blue color.
Comment/Suggestion:
[Discussion and Recommendations]
- I don’t fully understand your last recommendation (“The smartwatch synchronizes with a GPS satellite”). Could you consider reformulating this paragraph and making it more understandable?
- I would also suggest move Figure 27 from this section to another one.
- Moreover, in Discussion section, please also provide: study limitations, and implications for both theory and practice.
Answer:
According to the reviewer’s suggestion, the last recommendation has been clarified and detailed more exhaustively. Figure 27 was also unnecessary. For this reason, it has been removed to the manuscript. This section has been improved and extended to address the reviewer’s recommendations.
The principal limitation of this study was the lack of public documentation provided by Garmin regarding its API interface. In some cases, it was essential to study in deep other works from the forensic community. In addition to this, a forensic analysis of the communication carried out between the cloud services in charge of storing Garmin's activities and the user's device is offered, by proving that the security levels are the required to preserve sensitive data.
All changes included in the new version of the manuscript are marked in blue color.
Comment/Suggestion:
[Conclusions and Further Works] – please rename this section. “Conclusions” will be enough.
Frankly speaking, I want to thank you for reading such valuable study. Considering my perspective, I was comprehensibly informed. I think your study is interesting and timely.
Answer:
Thank you very much for your comments and suggestions to improve this study, as well as your great opinion about our work. All changes included in the new version of the manuscript are marked in blue color. We have also included this last change.
Reviewer 2 Report
This paper is a study of forensic analysis in model devices, especially the wearable sports equipment. With its popularity this becomes an important security issue for private risks of location, personal preference, etc. The major conclusion of this paper is a set of security guidelines based on a case study. There are some suggestions for edits below:
The information about popularity of the studied wearable (Garmin Forerunner Series) is missing. Although sports devices are popular, the prevalence of this specific brand (and mode) is unknown (people outside of Spain may not heard about it), which affect the importance of this question.
The paper contains lengthy screenshots about the data capture, but the explanation is to short in the process. The paper can be made more succinct if possible to move some into the appendix.
The Section 4 is just one case analysis, therefore not a research plan, which limits the extensibility of the conclusion.
Author Response
Comment/Suggestion:
This paper is a study of forensic analysis in model devices, especially the wearable sports equipment. With its popularity this becomes an important security issue for private risks of location, personal preference, etc. The major conclusion of this paper is a set of security guidelines based on a case study. There are some suggestions for edits below:
Answer:
Thank you very much for your comments and suggestions to improve the quality of our manuscript. All changes included in the new version of the manuscript are marked in blue color.
Comment/Suggestion:
The information about popularity of the studied wearable (Garmin Forerunner Series) is missing. Although sports devices are popular, the prevalence of this specific brand (and mode) is unknown (people outside of Spain may not heard about it), which affect the importance of this question.
Answer:
The manuscript has been revised to include a greater context of the research problem, not only in Spain. This way, additional references have been added with additional justifications. Smartwatches and fitness devices are the most popular due to their great mobility and connectivity capabilities. They are continuously connected to users’ mobile device with a large variety of sensors, and they offer notifications, alerts, recommendations, etc. This way, there has been a big increase in the number of physical activities carried out by people.
Additionally, more than 66% of runners use wearable devices to quantify their sport performance. Garmin has been found to be the most popular brand among users of sports devices with almost the 44% of runners, whereas Polar, TomTom, and Nike are used less. Garmin is one of the main companies in the sector. Runners are very motivated by documenting and tracking their activities, as well as supporting their goal-oriented reflections and actions.
All changes included in the new version of the manuscript are marked in blue color.
Comment/Suggestion:
The paper contains lengthy screenshots about the data capture, but the explanation is to short in the process. The paper can be made more succinct if possible to move some into the appendix.
Answer:
According to this suggestion, and others, some of the screenshots have been removed and/or moved to a new appendix section.
Comment/Suggestion:
The Section 4 is just one case analysis, therefore not a research plan, which limits the extensibility of the conclusion.
Answer:
Section 4 and the manuscript have been revised to clarify the research workflow, as well as justifying the focus of the work over specific Garmin smartwatches. Specifically, the emulation of a digital forensic analysis process and, also, the study of possible vulnerabilities among this Garmin device and its cloud services. From the data investigation process, several security guidelines and recommendations have been provided reader with the preservation of users’ data privacy.
All changes included in the new version of the manuscript are marked in blue color.
Round 2
Reviewer 1 Report
Dear Authors.
Thank you for addressing all my concerns. I do appreciate your hard work and motivation to make a better paper.
Since the first reading, my perception of your manuscript has been good, while now is very good. I am fully confident that this paper will bring a broad reading audience, as well as a considerable number of citations.
Author Response
Thank you very much for your comments and suggestions during the whole process, as well as your great opinion about our work.